# A non-canonical sensing pathway mediates *Plasmodium* adaptation to amino acid deficiency

Inês M. Marreiros [1,2], Sofia Marques [1], Ana Parreira[1], Vincent Mastrodomenico[3], Bryan C. Mounce [3,4], Chantal T. Harris[5,6], Björn F. Kafsack [5], Oliver Billker [7], Vanessa Zuzarte-Luís[1] & Maria M. Mota [1✉]

Eukaryotes have canonical pathways for responding to amino acid (AA) availability. Under AA-limiting conditions, the TOR complex is repressed, whereas the sensor kinase GCN2 is activated. While these pathways have been highly conserved throughout evolution, malaria parasites are a rare exception. Despite auxotrophic for most AA, *Plasmodium* does not have either a TOR complex nor the GCN2-downstream transcription factors. While Ile starvation has been shown to trigger eIF2α phosphorylation and a hibernation-like response, the overall mechanisms mediating detection and response to AA fluctuation in the absence of such pathways has remained elusive. Here we show that *Plasmodium* parasites rely on an efficient sensing pathway to respond to AA fluctuations. A phenotypic screen of kinase knockout mutant parasites identified nek4, eIK1 and eIK2—the last two clustering with the eukaryotic eIF2α kinases—as critical for *Plasmodium* to sense and respond to distinct AA-limiting conditions. Such AA-sensing pathway is temporally regulated at distinct life cycle stages, allowing parasites to actively fine-tune replication and development in response to AA availability. Collectively, our data disclose a set of heterogeneous responses to AA depletion in malaria parasites, mediated by a complex mechanism that is critical for modulating parasite growth and survival.

[1] Instituto de Medicina Molecular João Lobo Antunes, Faculdade de Medicina, Universidade de Lisboa, Lisboa, Portugal. [2] Instituto de Ciências Biomédicas Abel Salazar (ICBAS), Universidade do Porto, Porto, Portugal. [3] Department of Microbiology and Immunology, Stritch School of Medicine, Loyola University Chicago, Maywood, IL, USA. [4] Infectious Disease and Immunology Research Institute, Stritch School of Medicine, Loyola University Chicago, Maywood, IL, USA. [5] Department of Microbiology and Immunology, Weill Cornell Medical College, New York, NY, USA. [6] Immunology & Microbial Pathogenesis Graduate Program, Weill Cornell Medicine, New York, NY, USA. [7] Molecular Infection Medicine Sweden, Molecular Biology Department, Umeå University, Umeå S-90187, Sweden. ✉email: mmota@medicina.ulisboa.pt

Malaria, a disease caused by *Plasmodium* parasites, kills a child every minute. While there have been significant reductions in the number of infection and mortality rates since 2000 (51% and 27% reduction, respectively), the plateau in progress, first observed in 2017, still persists. Notably, during the past 2 years, the COVID-19 pandemic has resulted in unprecedent challenges to health systems worldwide, having disrupted malaria services, leading to a marked increase in cases and deaths. In 2020, there were an estimated 241 million malaria cases and 627,000 malaria deaths worldwide, which represents about 14 million more cases and 69,000 more deaths compared to 2019[1].

Both parasite and host genetics, as well as host immune responses, contribute to the outcome of infection[2–5]. Still, the most consistent indicator of disease severity is parasite biomass[6]. *Plasmodium* is a rapidly multiplying unicellular organism undergoing a complex developmental cycle that takes place in the mammalian and mosquito hosts; a lifestyle that requires rapid adaptation to distinct environments. In the mammalian host, *Plasmodium* initially infects hepatocytes to generate thousands of merozoites, which are later released into the bloodstream where they invade and replicate within red blood cells (RBCs). Each merozoite replicates by schizogony to generate 10–30 new merozoites every cycle, which occurs at 24, 48 or 72 h, depending on the *Plasmodium* species. As a rapidly multiplying obligatory intracellular parasite, *Plasmodium* has high nutritional demands relying on the nutrients provided by its host to survive. To cope with the different environments and conditions its life cycle entails, *Plasmodium* mainly relies on the expression of distinct transcriptomes at each developmental stage[7–9]. However, besides the ever-changing environment accompanying its normal life cycle progression, the environment within a single host is also extremely dynamic, with parasites constantly facing variable immune responses, drug pressures and metabolic fluctuations[10]. We and others have provided evidence that parasite adaptation to unpredictable fluctuations within each niche requires alternative strategies, such as efficient sensors, and a tight regulation of nutrient-sensing signaling pathways[11–13]. *Plasmodium* parasites are auxotrophic for most amino acids (AA), and the blood stages acquire the majority of them through the digestion of host erythrocyte hemoglobin[14,15]. However, some AA are either absent (isoleucine) or rare (methionine) within human hemoglobin and must be acquired from extracellular sources[16,17].

In eukaryotes, from yeast to mammals as well as plants, the target of the rapamycin (TOR) complex and the general control nonderepressible 2 (GCN2)/eIF2α signaling cascades are two well-characterized mechanisms to sense AA fluctuations[18,19]. Notably, no homolog of the TOR complex has been identified to date in *Plasmodium*; however, a downstream effector, *Maf1*, is conserved in *P. falciparum* and has been described as essential for the parasite's asexual blood stage[20]. On the other hand, although the *Plasmodium* eIK1 kinase[21] closely clusters with the GCN2 AA-sensing kinase, the parasite lacks some key components of this signaling pathway, including the downstream transcription factors as well as the biosynthetic pathways that mediate GCN2 action, namely the orthologues for starvation-response regulators such as GCN4 in yeast and ATF4 in mammals[22,23]. This has led some to propose that *Plasmodium* evolved a stripped-down starvation-response pathway[24], suggesting a reduced capacity to adapt to AA depletion, rather than actively responding to alterations in AA levels. The present paradigm is that *P. falciparum*, in the absence of canonical eukaryotic nutrient stress-response pathways, can cope with an inconsistent AA supply by hibernating until more nutrients are provided[24]. We now challenge this paradigm by showing that both *P. berghei* and *P. falciparum* parasites can actively mount a response to a decrease in two distinct AAs (methionine and isoleucine) by activating, in each life cycle stage, a different protein kinase, ultimately leading to a reprograming of replication and parasite development.

## Results

### AA depletion impacts *Plasmodium* development and replication rates.

To assess the effect of AA depletion during parasite intra-erythrocytic development, we monitored the growth of both *P. falciparum* and *P. berghei* parasites cultured in media with or without methionine (Met) or isoleucine (Ile) (Fig. 1). Sorbitol-synchronized *P. falciparum* NF54 wild-type (WT) parasites cultured in medium containing methionine (MetS) or isoleucine (IleS) (0.100 mM Met; 0.38 mM Ile) develop and replicate every 48 h, resulting in a marked increase in parasitemia with each generation (Fig. 1a). However, when cultured in either Met deficient (MetD) or Ile deficient (IleD) medium, parasites undergo impaired growth, as evidenced by a marked reduction in parasitemia (Fig. 1a). Notably, while the effect of IleD on parasitemia is evident from day 2 onwards, parasitemia in MetD only becomes significantly different from day 4 onwards. Such difference may be explained by the fact that Ile is totally absent from human hemoglobin, while Met is present, although at low levels[25]. Neither IleD[24] nor MetD impact parasite viability for a 48 h period, as synchronized ring-stage forms quickly resume growth upon AA supplementation (Supplementary Fig. 1a).

Methionine, among the twenty amino acids, plays an integral role in protein biosynthesis and in regulation of translation and global gene expression—as it is coded by the translation initiation codon. Met is metabolized into S-adenosylmethionine (SAM), the cellular methylation currency, via an ATP-dependent process which is catalyzed by the *Plasmodium* SAMS enzyme. Therefore, besides MetD we have also focused on the SAMS enzyme, since SAM is the major biological methyl donor and also the precursor of polyamines via the aminopropylation pathway, or glutathione (GSH), upon entry into the transsulfuration pathway, where homocysteine (Hcy) is metabolized to GSH (Fig. 2a). Addressing both MetD and parasites knockdown for the SAMS enzyme allowed us to establish the role of Met in protein incorporation but also to disentangle the role of methylation, GSH and polyamines in parasite growth. To do so, we employed a previously generated[26] glucosamine (GlcN)-regulatable parasite line in which the *P. falciparum* SAMS, the first and rate-limiting enzyme of the methionine cycle, was conditionally knocked down (*Pfsams-glmS* parasite line). Interestingly, the knockdown of the SAMS enzyme, achieved by GlcN addition to the culture media (*Pfsams*-glmS +GlcN), phenocopies WT parasites growing in MetD media, as evidenced by a marked reduction in parasitemia from day 4 onwards (Fig. 1a).

We next sought to determine whether reduced parasitemia observed upon AA limitation was an outcome of defective parasite development. We employed the *P. falciparum* 3D7 strain which exhibited a shorter life cycle length, replicating every 36 h to 42 h in control conditions, in accordance to previous studies estimating a 38.8 h cell cycle length for this particular strain[27,28]. Time course analysis of synchronized *P. falciparum* 3D7 ring-stages growing in MetD or IleD media reveals impaired parasite development in both conditions, yet with some particularities unique to each condition (Supplementary Figs. 1b, c and 2). Both MetD- and IleD- media growing parasites fail to complete maturation in a timely and coordinated manner, arresting at the immature schizont-stage, as evidenced by an increase in the time spent at this stage. Ile depletion, however, has a more pronounced impact on cell cycle progression, affecting also the trophozoite stage, with parasites arresting for one complete developmental cycle at this particular stage (Supplementary Fig. 1c). In fact, such

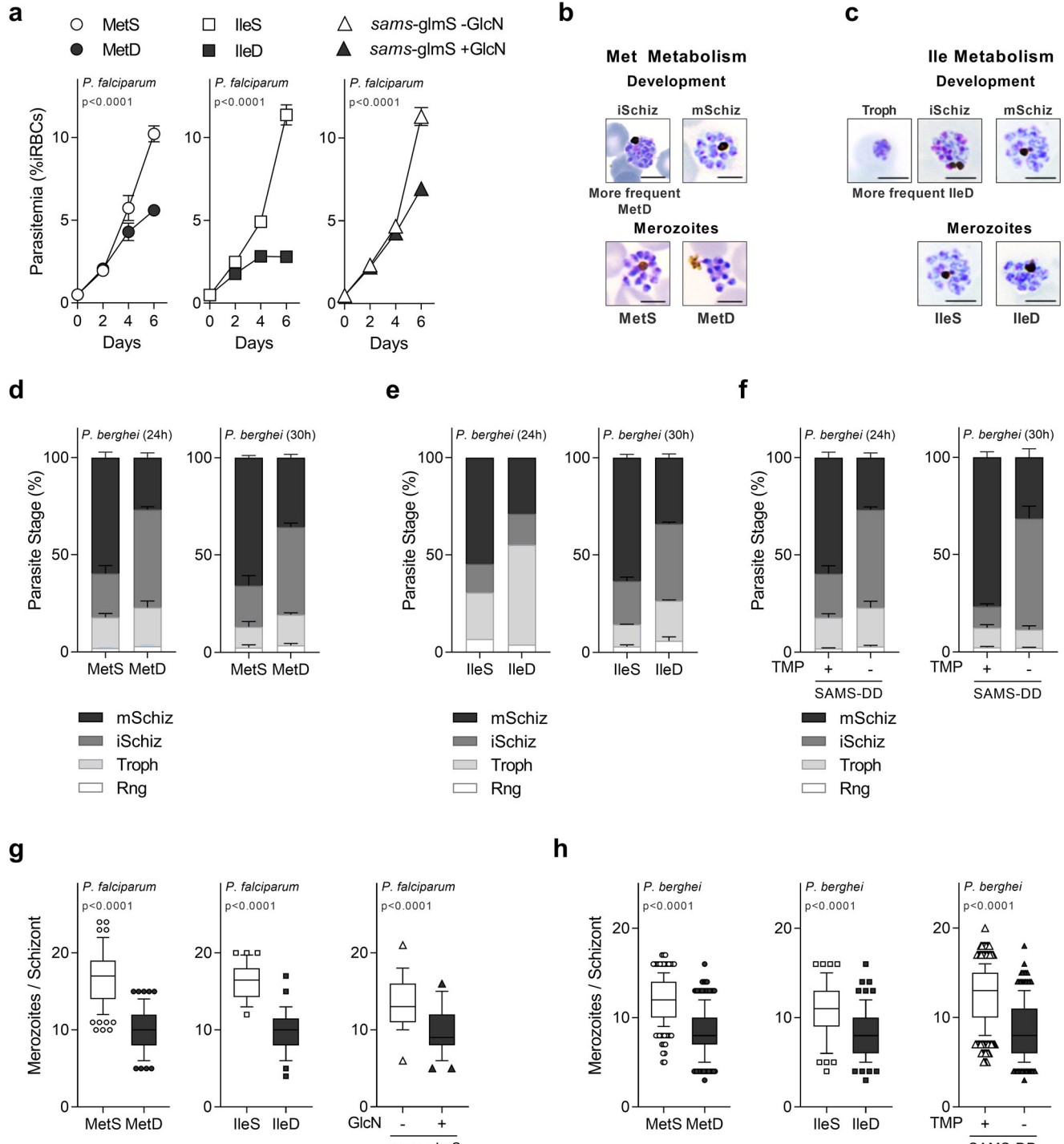

**Fig. 1 AA availability impacts parasite intra-erythrocytic development and replication.** Parasitemia of synchronized *P. falciparum* wt parasites cultured in MetD or IleD media and *P. falciparum* parasites knockdown for the SAMS enzyme (*Pfsams*-glmS +GlcN). **b**, **c** Representation of *P. berghei* ex vivo development and replication in (**b**) MetD or (**c**) IleD media for 30 h. Scale bar = 5 µm. **d**–**f** Quantification of *P. berghei* intra-erythrocytic developmental stages in (**d**) MetD or (**e**) IleD media and in (**f**) parasites knockdown for the SAMS enzyme (*Pb*SAMS-DD -TMP) after ex vivo cultivation for 24 h or 30 h. **g**, **h** Mean merozoite number per schizont in (**g**) *P. falciparum* or (**h**) *P. berghei* parasites, cultured in MetD or IleD media or in parasites knockdown for the SAMS enzyme. **a**. Data represents the mean percentage of iRBCs ±SEM (2-way ANOVA); n = 2 independent experiments. **d**–**f** Data is shown as mean percentage of parasites at each developmental stage with error bars representing SEM; n = 3 independent experiments, except for Ile at 24 h (n = 2). Individual data points are provided in Supplementary Fig. 3. Rng ring, troph trophozoite, iSchiz immature schizont, mSchiz mature schizont. **g**, **h** Data is represented as box-whisker plot (10–90 percentile) of mean merozoite number per schizont ±SD (Mann–Whitney) in 2 independent experiments, except in **h**. *Pb*MetS, MetD (n = 3) and *Pb*SAMS-DD + / − TMP (n = 5) and for the following number of schizonts (N): **g** MetS, N = 107, MetD, N = 86; IleS, N = 32, IleD, N = 37; *sams*-glmS -GlcN, N = 25; *sams*-glmS + GlcN, N = 45; **h** MetS, N = 235; MetD, N = 275; IleS, N = 65, IleD, N = 70; SAMS-DD + TMP, N = 405, SAMS-DD -TMP, N = 325.

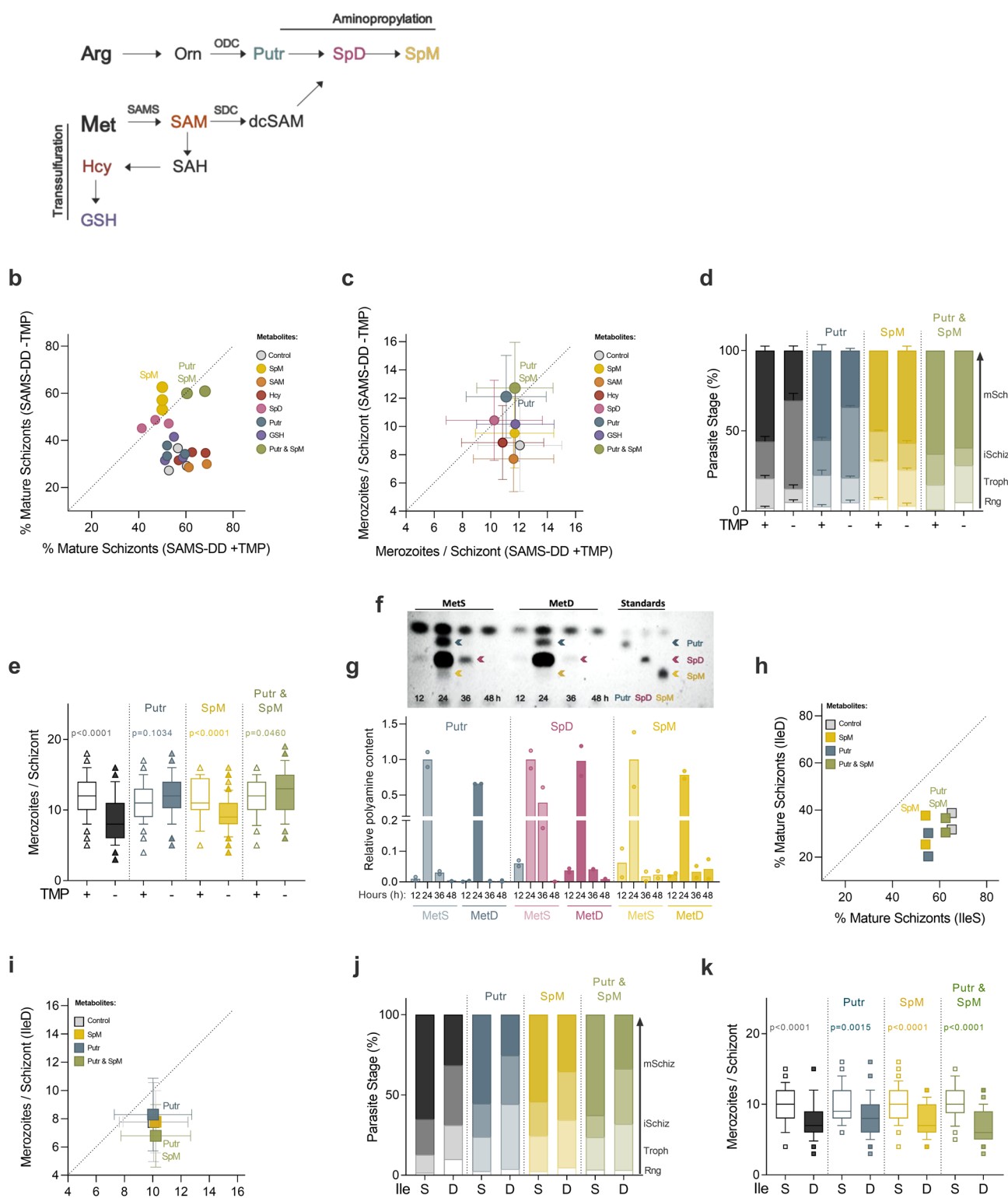

a remarkable delay in progression of isoleucine-starved parasites through the trophozoite stage was first documented by Babbit et al., who proposed a dormancy-like model for AA starvation[24].

This immediate and pronounced effect of Ile depletion on parasite growth, compared to methionine depletion, could result from a faster depletion of Ile stocks, as it is totally absent from human Hb. However, our results show that even though the

appearance of next generation merozoites is delayed both in MetD and IleD conditions, parasites can invade erythrocytes efficiently, as evidenced by the emergence of second-generation ring-stage forms that peak at ~72–84 h of development (Supplementary Fig. 1b, c).

Notably, when employing a malaria parasite of rodents, *P. berghei* ANKA, a similar impact was observed in MetD and IleD

**Fig. 2 Polyamines regulate parasite response to methionine depletion. a** Schematic representation of the close connection between the methionine cycle and the aminopropylation pathway in malaria parasites, both pathways relying on the *Plasmodium* bifunctional SDC/ODC enzyme. **b, c** Percentage of mature segmented schizonts and (**c**) merozoite numbers per schizont in *Pb*SAMS-DD + TMP *vs Pb*SAMS-DD -TMP (knockdown) parasites cultured with the Met-downstream metabolites. **d, e** Quantification of (**d**) *P. berghei* intra-erythrocytic developmental stages and (**e**) merozoite numbers in *P. berghei* parasites knockdown for the SAMS enzyme after supplementation with the polyamines putrescine (Putr), spermine (SpM) or both (Putr & SpM). **f, g** Polyamine visualization by thin layer chromatography (TLC) analysis, representative of *n* = 2 independent experiments, in synchronized *P. falciparum* NF54 WT parasites cultured in MetD media. Each lane represents a different timepoint. **g** Quantification of relative polyamine levels at 12 h (ring-stage), 24 h (trophozoite-stage), 36 h (schizont-stage) and 48 h (next generation ring-stages) after sorbitol synchronization in MetS and MetD conditions. The same number of parasites was loaded per condition and timepoint. Due to its very low abundance in the parasite, SpM could only be detected by TLC analysis 24 h after sorbitol synchronization. Blue, pink and yellow arrows represent Putr, spermidine (SpD) and SpM, respectively, in standards and samples. **h, i** Percentage of mature segmented schizonts and (**i**) mean merozoite numbers in *P. berghei* WT parasites cultured in IleS *vs* IleD media supplemented with the polyamines Putr, SpM or both. **j, k** Quantification of (**j**) *P. berghei* intra-erythrocytic developmental stages and (**k**) mean merozoite numbers in *P. berghei* WT parasites cultured in IleS or IleD media supplemented with polyamines. **b, c** Data represents the (**b**) percentage of mature schizonts and the **c** mean merozoite numbers produced ex vivo by *Pb*SAMS-DD -TMP (knockdown) parasites after supplementation with the Met-downstream metabolites. Polyamine supplementation was examined in *n* = 3 independent experiments, except for Putr +SpM and SAM (*n* = 2). Error bars represent the SD. **d, j** Data is shown as mean percentage of parasites at each developmental stage, with error bars representing SEM. Parasite staging was performed in triplicate, averaged and repeated in **d** *n* = 3, except for Putr +SpM (*n* = 2) or **j** *n* = 2 independent experiments. Individual data points are represented in Supplementary Fig. 6a, b. Rng ring, troph trophozoite, iSchiz immature schizont, mSchiz mature schizont. **e, k** Data is represented as box-whisker plot (10–90 percentile) of mean merozoite number per schizont ±SD (2-way ANOVA). Boxplots show the data of *n* = 2 independent experiments and for the following number of schizonts (*N*): **e** SAMS-DD + TMP, *N* = 122, -TMP, *N* = 117; +TMP + Putr, *N* = 50; -TMP + Putr, *N* = 48; +TMP + SpM, *N* = 37; -TMP + SpM, *N* = 102; +TMP + Putr & SpM, *N* = 47; -TMP + Putr & SpM, *N* = 47; **k** IleS, *N* = 38; IleD, *N* = 47; IleS +Putr, *N* = 59; IleD +Putr, *N* = 66; IleS +SpM, *N* = 56; IleD + SpM, *N* = 57; IleS +Putr & SpM, *N* = 38, IleD +Putr & SpM, *N* = 53. **g** Data represents the relative polyamine levels, normalized to the 24 h timepoint. *n* = 2 independent experiments**. h, i** Data represents the (**h**) percentage of mature schizonts and the (**i**) mean merozoite numbers produced ex vivo by *Pb* wt parasites growing in IleD media supplemented with polyamines. Polyamine supplementation was examined in *n* = 2 independent experiments. Error bars represent the SD.

conditions (Fig. 1b, c). Blood stages of rodent malaria parasites can only be maintained in culture for one developmental cycle as newly developed merozoites cannot reinvade erythrocytes anew. Even though synchronization cannot be achieved in these short-term ex vivo systems, parasites fully mature from rings/young trophozoites to segmented schizonts that do not rupture, even when fully mature[29]—allowing for a thorough characterization of schizont development. Microscopy analysis show that *P. berghei* parasites either growing in MetD or IleD media exhibit impaired development ex vivo, with only 30% of the parasites successfully developing into mature schizonts (Fig. 1d, e and Supplementary Fig. 3a, b). As for *P. falciparum*, while IleD leads to a significant delay in both trophozoite- and immature- schizont stages, MetD mainly impacts the immature-to-mature schizont transition. Surprisingly, providing *P. berghei* parasites an additional 6 h of maturation time (30 h timepoint) did not reflect in increased maturation rates under AA deficiency (Fig. 1d, e). This suggests that ~70% of parasites growing under AA-limiting conditions exhibit growth arrest, resembling the hibernation phenotype previously described by Babbit et al.[24]. Again, a *P. berghei* transgenic parasite line (*Pb*SAMS-DD parasite line) in which the first enzyme of the methionine cycle was knocked down (*Pb*SAMS-DD -TMP) phenocopied WT parasites cultured in MetD conditions (Fig. 1f and Supplementary Fig. 3c). In this species, SAMS knockdown was achieved by employing the destabilizing domain (DD) system[30]. The construct containing the *sams* gene fused to the DD and the hemagglutinin (HA) tag was introduced into *P. berghei* genome using double crossover homologous recombination (Supplementary Fig. 1d, e). Protein stabilization in vivo was achieved by providing trimethoprim (TMP) in drinking water for 2 days prior to infection (*Pb*SAMS-DD + TMP), while SAMS conditional knockdown (*Pb*SAMS-DD -TMP) was achieved in non-TMP treated mice, as evidenced by a significant decrease in SAMS-HA-DD protein levels, measured by immunoblotting analysis (Supplementary Fig. 1f). Similar results were observed by immunofluorescence analysis of *P. berghei* trophozoites and schizonts in which the SAMS enzyme was knocked down (*Pb*SAMS-DD -TMP; Troph, Schiz), as evidenced by a significant decrease in SAMS-HA-DD protein levels, relative

to control parasites (Supplementary Fig. 1g). Notably, *Pb*SAMS-DD -TMP merozoites exhibit SAMS expression levels comparable to those observed in control parasites (Supplementary Fig. 1h).

*Plasmodium* schizogony culminates in the production of individualized merozoites that quickly re-invade new erythrocytes. We observed that under AA deficient conditions, ~30% of parasites successfully complete the cell cycle in a timely manner, developing into mature schizonts containing individualized next generation merozoites. Yet, the number of merozoites each schizont contained was dependent on the growth conditions. Microscopy and flow cytometry analysis in *P. falciparum* and *P. berghei* showed significantly fewer merozoites in schizonts grown in MetD or IleD media, as well as in transgenic parasites knockdown for the SAMS enzyme (*Pfsams-glmS* + GlcN; *Pb*SAMS-DD -TMP) (Fig. 1g, h and Supplementary Fig. 4a–d). Allowing *P. berghei* parasites an additional 6 h maturation time (30 h of culture) did not restore merozoite numbers under either condition (Supplementary Fig. 4e), suggesting that lower replication rates result neither from the asynchronous nature of *Plasmodium* schizogony nor from delayed maturation, but rather from the ability of parasites to adapt to low AA levels.

To disclose the role of AA metabolism in the dynamics of an in vivo malaria infection, we provided a short-term, methionine-deprived (0% methionine, MetD) regimen to BALB/c mice. Body weight analysis shows that, in accordance to previous findings[31], MetD regimen leads to weight loss relative to the methionine-sufficient (MetS) regimen (Supplementary Fig. 4f). To exclude severe weight loss as a confounder, the MetD regimen was initiated 3 weeks before *P. berghei* ANKA infection and mice were only infected when their body weight had stabilized. The 3-week time point was chosen based on previous studies showing steady body weight and an improved metabolic profile after a 2–3-week regimen switch from *ad libitum* to calorie restriction[12]. Flow cytometry analysis of MetD-fed BALB/c mice infected with WT parasites shows a significant reduction in peripheral parasitemia relative to the MetS-fed group (Supplementary Fig. 4g). Interestingly, the same was observed in mice fed on a MetS diet but infected with *Pb*SAMS-DD -TMP parasites (Supplementary Fig. 4h). Moreover, the reduced number of merozoites per

schizont we observed after conditional knockdown of *Pb*SAMS ex vivo was also confirmed in vivo (Supplementary Fig. 4i). Altogether, these findings show that fluctuations in AA availability have an effect on parasite growth, by either stalling progression through the intra-erythrocytic asexual cell cycle, via a mechanism resembling hibernation, or by reducing parasite replication rates.

**Polyamines regulate *Plasmodium* response to methionine depletion.** Amino acids are not only the building blocks of proteins but also serve as precursors for the synthesis of many metabolites with diverse functions in growth and other biological processes of a living organism. It is therefore likely that disruption of AA metabolism will profoundly affect the levels of many other AA-dependent metabolites. While Ile shares the first enzymatic steps of its metabolism with two other AAs (leucine and valine)[32], the initial steps of Met metabolism are unique among all AAs, rendering the products of its metabolism ideal candidates. To find out which Met-downstream metabolites could rescue attenuated replication and impaired development in *P. berghei* parasites knockdown for the SAMS enzyme, we supplemented media with SAM, Hcy, GSH or the polyamines putrescine (Putr), spermidine (SpD) and spermine (SpM) (Fig. 2a).

Microscopic analysis shows that SAM rescues neither impaired maturation nor reduced numbers of merozoites produced by *Pb*SAMS-DD -TMP schizonts, relative to the control group (Fig. 2b, c and Supplementary Fig. 5). This is not surprising since SAM transport across the plasma membrane occurs to a minimal extent in mammalian cells[33]. However, the results show that among the SAM-downstream metabolites, the polyamines Putr and SpM are the only metabolites governing parasite's response to SAMS deficiency (Fig. 2b–e and Supplementary Fig. 6a). In schizonts lacking the SAMS enzyme (*Pb*SAMS-DD -TMP), while SpM reverses impaired maturation, Putr fully restores merozoite numbers (Fig. 2b–e). Furthermore, the simultaneous addition of SpD and Putr (Putr & SpD) rescues not only the impaired development of *Pb*SAMS-DD -TMP parasites but also the reduced replication rates, which is reflected in a higher percentage of mature schizonts and an increased number of merozoites per schizont (Fig. 2b–e). These data support the existence of two distinct parasite populations, one that stalls development and hibernates and other that, while completing the cycle at a normal pace, develops into schizonts containing fewer offspring. These two parasite populations are independently regulated by two distinct metabolites (SpM or Putr) and are likely governed by different mechanisms.

In an attempt to unveil how parasite replication and development are independently regulated by distinct polyamines, we assessed the impact of MetD on polyamine biosynthesis throughout parasite intra-erythrocytic development. Thin layer chromatography (TLC) analysis of *P. falciparum* NF54 parasites shows an overall reduction in polyamine synthesis throughout development in MetD (Fig. 2f, g and Supplementary Fig. 11). Notably, the effect of MetD on Putr production is more prominent in early stages of parasite development (trophozoite, 24 h) whereas the production of SpD (and likely that of its downstream product, SpM) is mainly reduced during late schizogony (36 h). Due to its very low abundance in the parasite, SpM levels in TLC analysis were significantly lower than those observed for Putr and SpD (Fig. 2f). This supports our previous findings in the rodent malaria species, *P. berghei* ANKA, showing a role for Putr during replication (Fig. 2e), and for SpM at later stages of parasite development, particularly during schizont segmentation (Fig. 2d). These findings were recapitulated in the human malaria parasite, *P. falciparum*, in MetD conditions, as

evidenced by a sharp increase in parasitemia upon polyamine supplementation (Supplementary Fig. 6c).

Indeed, our data are consistent with previous findings showing that inhibition of *P. falciparum* ODC enzyme (critical for Putr synthesis) induces cytostatic arrest at the trophozoite stage and halts parasite development at G1/S transition, a phenotype that is rescued by the addition of exogenous polyamines[34–36]. So far, our data show that the developmental and replication responses to methionine depletion are independent and each rely on a different polyamine. Not surprisingly, the effects of Met-downstream metabolites, namely polyamines, did not extend to conditions of Ile depletion, showing that these were specific to Met depletion signaling (Fig. 2h–k and Supplementary Fig. 6b). The data show that when each amino acid (Met or Ile) is depleted, separate mechanisms are orchestrated, with the downstream metabolites triggering these responses being specific to the metabolic pathway that is affected. It is important to note that the different responses observed at the phenotypic level likely translate to the different roles each AA plays in the cell. Our findings so far led us to hypothesize that *Plasmodium* parasites have the ability to sense fluctuations in AA levels—as well as in the downstream metabolites—which independently regulate parasite development or replication, functioning as key molecular players during distinct stages of the parasite's life cycle.

**Nek4, eIK1 and eIK2 are key regulators of parasite response to AA limitation.** Cells respond to environmental stress conditions by adjusting gene expression and protein synthesis, a role mainly filled by protein kinases signaling pathways. However, the key molecular players involved in AA-sensing regulation in other organisms are absent from the genome of the malaria parasite. To identify potential regulators, we examined a panel of 15 eukaryotic protein kinases (ePKs) which were previously shown to be non-essential for *P. berghei* asexual erythrocytic development[37]. By growing *P. berghei* lines with loss-of-function mutations in protein kinase genes[37] in AA depleted media (Fig. 3, Supplementary Figs. 7, 8) we have identified Nek4, eIK1 and eIK2 mutants as unresponsive to both MetD and IleD, suggesting a role for these kinases in AA sensing, albeit with some relevant particularities at the phenotypic level (Fig. 3 and Supplementary Fig. 9).

In MetD or IleD conditions, Δnek4 parasites mature and replicate similarly to wild-type parasites in MetS or IleS (Fig. 3). In contrast, while ΔeIK1 parasites in MetD or IleD conditions mature to rates comparable to wild-type parasites in control media, the merozoite numbers are still markedly diminished. Interestingly, ΔeIK2 parasites exhibit the opposite response under MetD or IleD conditions, with maturation rates being diminished, yet merozoite numbers generated per schizont resemble wild-type parasites growing in control conditions (Fig. 3). When assessing parasite growth, replication rates and intra-erythrocytic development (Fig. 4a–d and Supplementary Fig. 10), *P. falciparum* kinase mutant knockout lines (eIK1[21], eIk2[38] and nek4[39]) showed a similar response to MetD or IleD conditions. Interestingly, the inability of kinase mutant knockout parasites to control growth in response to AA depletion likely leads to parasite death after several rounds of replication, as illustrated by a stalling (ΔeIK2, ΔeIK1) or even decline (Δnek4) in parasitemia from 96 h onwards (Fig. 4a, b). These findings show that the previously described kinases function as a coping strategy for Met or Ile shortage, as parasites lacking these specific kinases cannot fine-tune growth to nutrient availability and thus fail to enter hibernation and to adjust replication rates.

The involvement of Nek4, eIK1 and eIK2 in parasite adaptation to AA deficiency strongly suggests that these kinases regulate an

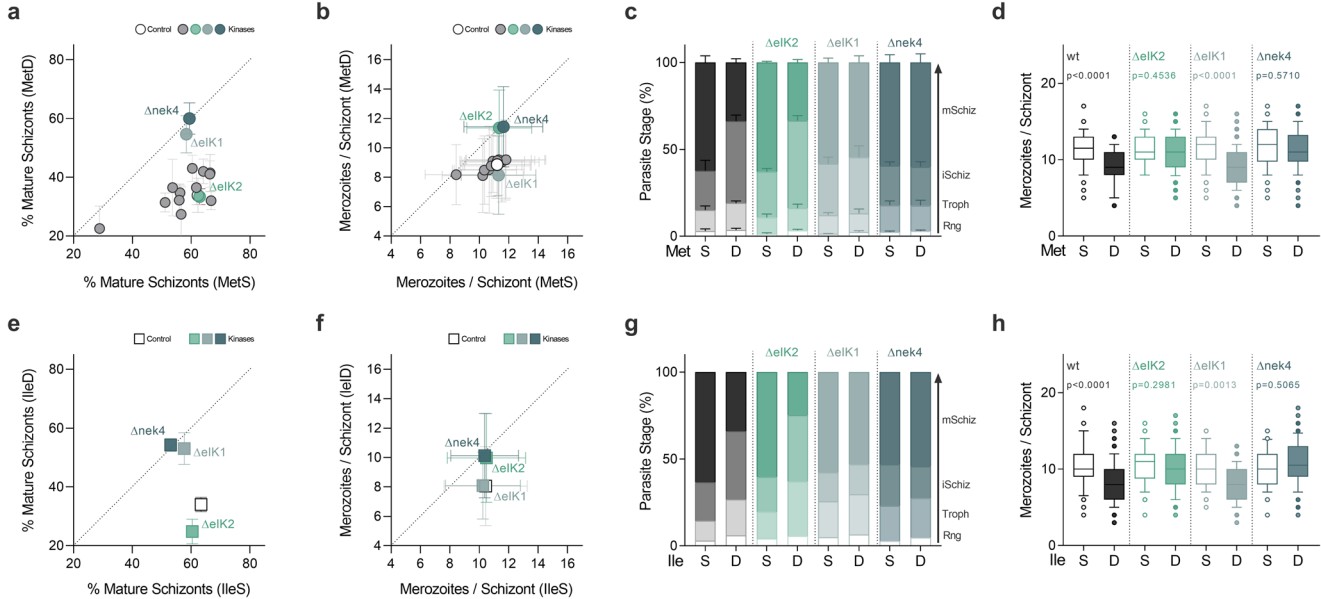

**Fig. 3 A signaling pathway comprising 3 distinct kinases regulates *Plasmodium berghei* stress response to AA depletion. a, b** Screen of 15 non-essential *P. berghei* kinases in loss-of-function mutants using the ex vivo maturation assay. **a** Percentage of mature segmented schizonts and (**b**) merozoite numbers per schizont in wild-type and kinase knockout lines (Δnek4: PBANKA_061670; ΔeIK1: PBANKA_1308400; ΔeIK2: PBANKA_0205800) cultured in MetS *vs* MetD media. **c, d** Quantification of (**c**) *P. berghei* intra-erythrocytic developmental stages and (**d**) merozoite numbers of *P. berghei* wild-type and kinase mutant ΔeIK2, ΔeIK1 and Δnek4 parasites cultured in MetS or MetD media. **e, f** Percentage of mature segmented schizonts and (**f**) merozoite numbers in wild-type and kinase knockout parasite lines cultured in IleS *vs* IleD media. **g, h** Quantification of (**g**) *P. berghei* intra-erythrocytic developmental stages and (**h**) merozoite numbers of *P. berghei* wild-type and kinase mutant ΔeIK2, ΔeIK1 and Δnek4 parasites cultured in IleS or IleD media. **a, b, e, f** Kinase knockout mutants were analyzed in triplicate and averaged, with each data point representing the **a, e** mean percentage of mature schizonts or **b, f** the mean merozoite number produced per schizont. Error bars represent the SE Diff or SD between replicates, respectively. $n = 3$ independent experiments for wild-type, ΔeIK2, ΔeIK1 and Δnek4 or $n = 2$ for the other knockout lines. **c, g** Parasite staging was performed in triplicate, averaged and repeated in **c**. $n = 3$ or **g**. $n = 2$ independent experiments. Data is shown as mean percentage of parasites at each developmental stage, with error bars representing SEM. Individual data points are represented in Supplementary Fig. 9a, b. **d, h** Data is represented as box-whisker plot (10–90 percentile) of mean merozoite number per schizont ±SD (2-way ANOVA). Boxplots show the data of **d** $n = 3$ or **h** $n = 2$ independent experiments and for the following number of schizonts (N), wild-type: MetS, $N = 148$; MetD, $N = 148$; IleS, $N = 114$; IleD, $N = 133$; ΔeIK2: MetS, $N = 85$; MetD, $N = 158$; IleS, $N = 70$; IleD, $N = 68$; ΔeIK1: MetS, $N = 220$; MetD, $N = 238$; IleS, $N = 75$; IleD, $N = 94$ and Δnek4: MetS, $N = 106$; MetD, $N = 106$; IleS, $N = 90$; IleD, $N = 102$.

active parasite response to AA starvation, whose modulation occurs in a stage-specific and temporally-regulated manner, a mechanism that, to the best of our knowledge, is hitherto unknown. Moreover, our data strongly suggests that Nek4 acts upstream to eIK1 and eIK2 in the pathway, regulating both replication (via eIK2) or development (via eIK1).

**Parasite response to low AA implies eIF2α phosphorylation at different stages of the cell cycle.** *P. falciparum* parasites growing in AA-free media exhibit increased levels of phosphorylation of the eukaryotic initiation factor 2α (eIF2α)[21], which is a well characterized mechanism in eukaryotic model organisms that blocks translation under AA depletion[40–42]. From yeast[43] to mammals[44], such a mechanism is coordinated by the eIF2α kinase GCN2, which recognizes the increased accumulation of uncharged tRNAs induced by AA depletion[41,42]. Three eIF2α kinases have been identified in *Plasmodium*: eIK1, eIK2 and PK4[45] whose only described function is to phosphorylate the *Plasmodium* eIF2α orthologue under different stresses[28]. Since our screening data revealed a key role for eIK1 and eIK2 (two of the three eIF2α kinases) in parasite response to AA depletion, we next sought to establish whether these kinases coordinate an adaptive response mechanism and thereby modulate the phosphorylation of their downstream target eIF2α.

Immunoblotting analysis of synchronized *P. falciparum* WT parasites reveals that under control conditions (MetS, IleS conditions) eIF2α phosphorylation is lower at the ring stage,

increasing later during schizont development (Fig. 5 and Supplementary Fig. 12), a process required for successful schizont maturation, through reduction of protein synthesis[46]. By contrast, MetD and IleD conditions impact on eIF2α phosphorylation status (Fig. 5a, b), suggestive of an early and active parasite response to low AA levels, yet with some particularities—as previously observed at the phenotypic level (Fig. 1). Under both MetD and IleD conditions, eIF2α phosphorylation is strongly induced early in development, at the ring stage. However, while enhanced phosphorylation is maintained throughout schizont development in MetD conditions (Fig. 5a); under IleD, eIF2α phosphorylation is rapidly lost and restored to basal levels as schizont development progresses (Fig. 5b). This is consistent with our previous findings that IleD primarily affects early developmental stages, while MetD has the greatest impact during schizont development (Supplementary Fig. 1b, c).

Our results show that ΔeIK1 parasites can only respond to AA depletion during the schizont stage, as evidenced by increased eIF2α phosphorylation levels at this particular stage and an apparent loss of eIF2α phosphorylation during ring-stage development (Fig. 5a, b). By contrast, while ΔeIK2 schizonts are incapable of phosphorylating eIF2α, ring-stage parasites successfully respond and adapt to AA scarcity. Remarkably, Δnek4 parasites are completely unable to respond to MetD, with negligible eIF2α phosphorylation levels in both the ring and schizont stages (Fig. 5a). Therefore, while eIK1 seems to be critical for parasite's response during early-stages of development,

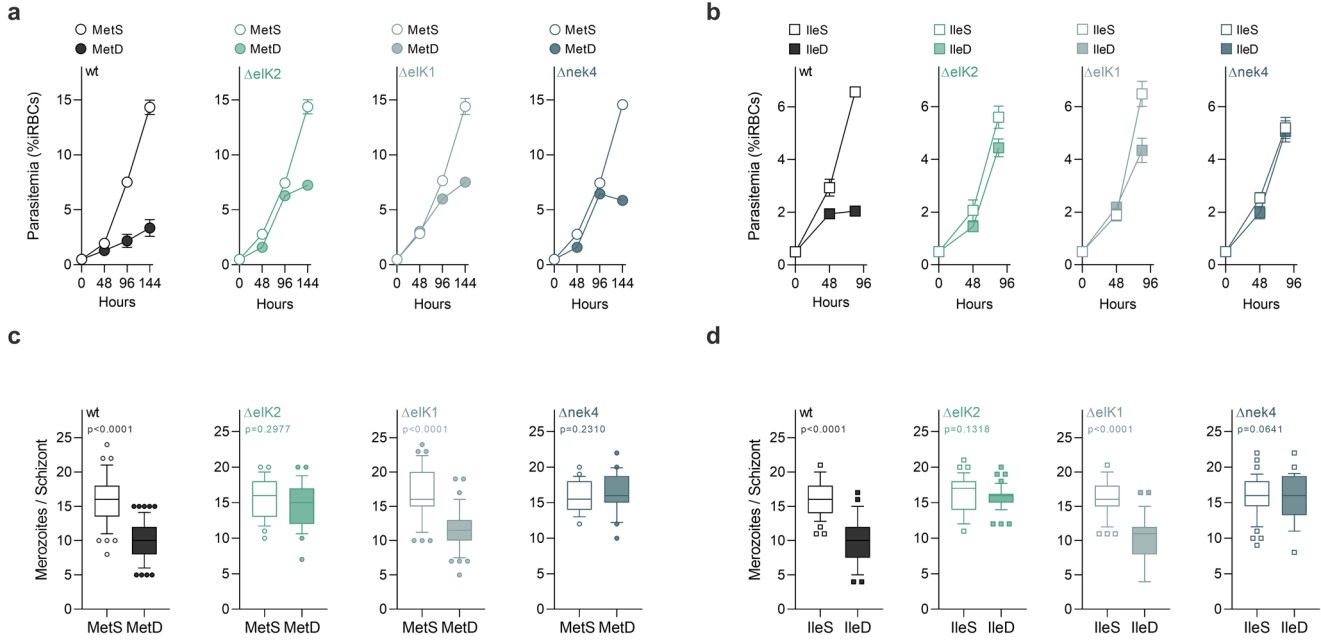

**Fig. 4 elk1, elk2 and Nek4 function is conserved in the human malaria parasite *P. falciparum*. a, b** Parasitemia of synchronized *P. falciparum* 3D7 WT, ΔeIK2 (PF3D7_0107600), ΔeIK1 (PF3D7_1444500) and Δnek4 (PF3D7_0719200) parasites cultured in (**a**) MetS or MetD media and (**b**) IleS or IleD media. **c, d** Merozoite numbers per schizont in *P. falciparum* parasites cultured in (**c**) MetS or MetD media and (**d**) IleS or IleD media. **a, b**. Data represents the mean percentage of iRBCs. Parasitemia was performed in triplicate, averaged and repeated in *n* = 2 independent experiments. **c, d** Data is represented as box-whisker plot (10–90 percentile) of mean merozoite number per schizont ± SD (Mann-Whitney). Boxplots show the data of *n* = 2 independent experiments and for the following number of schizonts (*N*): wild-type, MetS, *N* = 57; MetD, *N* = 58; IleS, *N* = 37; IleD, *N* = 33; ΔeIK2, MetS, *N* = 26; MetD, *N* = 25; IleS, *N* = 37; IleD, *N* = 42; ΔeIK1, MetS, *N* = 35; MetD, *N* = 44; IleS, *N* = 41; IleD, *N* = 36; and Δnek4, MetS, *N* = 22; MetD, *N* = 20; IleS, *N* = 45; IleD, *N* = 28.

eIK2 activity emerges as only relevant towards late stages of development—at least in MetD. Remarkably, Nek4 activity appears to be critical in the parasite's response to AA deficiency throughout asexual intraerythrocytic development (Fig. 5a)—a kinase previously known to function during gametocytogenesis[47]. While it has previously been shown that ring-stage parasites lacking eIK1 do not phosphorylate eIF2α in response to AA depletion, the ability of schizonts to actively respond to AA fluctuations, as well as the role of eIK2 (whose only known function is in the sporozoite stage[21]) and Nek4 in response to AA depletion remained unknown.

Overall, the data show that *Plasmodium* parasites can detect AA fluctuations and mount active responses to individual AAs. This occurs via a common mechanism comprising the phosphorylation of eIF2α by two different kinases—eIK1 and eIK2—whose function is likely coordinated by Nek4 in a temporally-regulated manner, with each kinase affecting a specific, but sequential, stage of development (Fig. 5c).

## Discussion

Nutrient scarcity is an important selective pressure that has shaped the evolution of many organisms and most cellular processes. While nutrient sensing pathways have been established and studied in model systems, malaria parasites' ability to sense nutrients has only recently been uncovered. This owes to the fact that *Plasmodium* parasites do not appear to possess any of the canonical eukaryotic nutrient-sensing pathways, including the SNF/AMPK, TOR and GCN2 biosynthetic pathways. Notably, while the canonical SNF/AMPK energy-sensing pathway was previously thought to be absent, we have recently shown that a parasite kinase, KIN, is a putative functional AMPK homolog, working as a broad metabolic/energy sensor that drives an active and coordinated parasite response to energy fluctuations[12]. On the

other hand, the master regulator of cell growth in eukaryotes (the TOR complex) is completely absent in *Plasmodium*, which has been shown to possess only a rudimentary AA starvation-sensing eukaryotic initiation factor 2α (eIF2α), that does not directly promote parasite survival upon AA starvation[21,24]. In many model organisms, GCN2 has been shown to sense the uncharged tRNAs that accumulate upon AA deprivation. GCN2 attenuates translation, which not only consumes AA, but is also one of the most energy-demanding cellular processes[41,42]. Our extensive phenotypic analysis in both MetD or IleD media and in the kinase knockout mutant parasite lines has allowed the identification of 3 distinct parasite kinases that function as key regulators of parasite growth under AA scarcity: Nek4, a kinase so far only described to function during gametocyte development[47], eIK1 and eIK2. The inability to reconcile growth with AA availability in the absence of such kinases suggests that asexual blood stage parasites can actively adapt to a decrease in AAs via these respective kinases. Additionally, our data suggests that these two distinct eIF2α kinases function in a temporally-regulated manner, with eIK1 regulating developmental stage transition but not replication and eIK2 showing the opposite behavior as it regulates replication but not development.

While the distinct dynamics of eIF2α phosphorylation in each kinase mutant during development under AA-deprived conditions supports this, additional research into kinase activity levels and activation status at each parasite stage would corroborate such a complex model. Indeed, our data suggest that in order to survive under AA deficiency, malaria parasites must be able to modulate replication levels or enter hibernation. Thus, decreased replication and halted cell cycle progression in wild-type parasites mirrors the ability of malaria parasites to successfully respond to AA depletion; an adaptive mechanism that requires enhanced phosphorylation of eIF2α, which must either be maintained throughout development (MetD) or not (IleD), depending on the

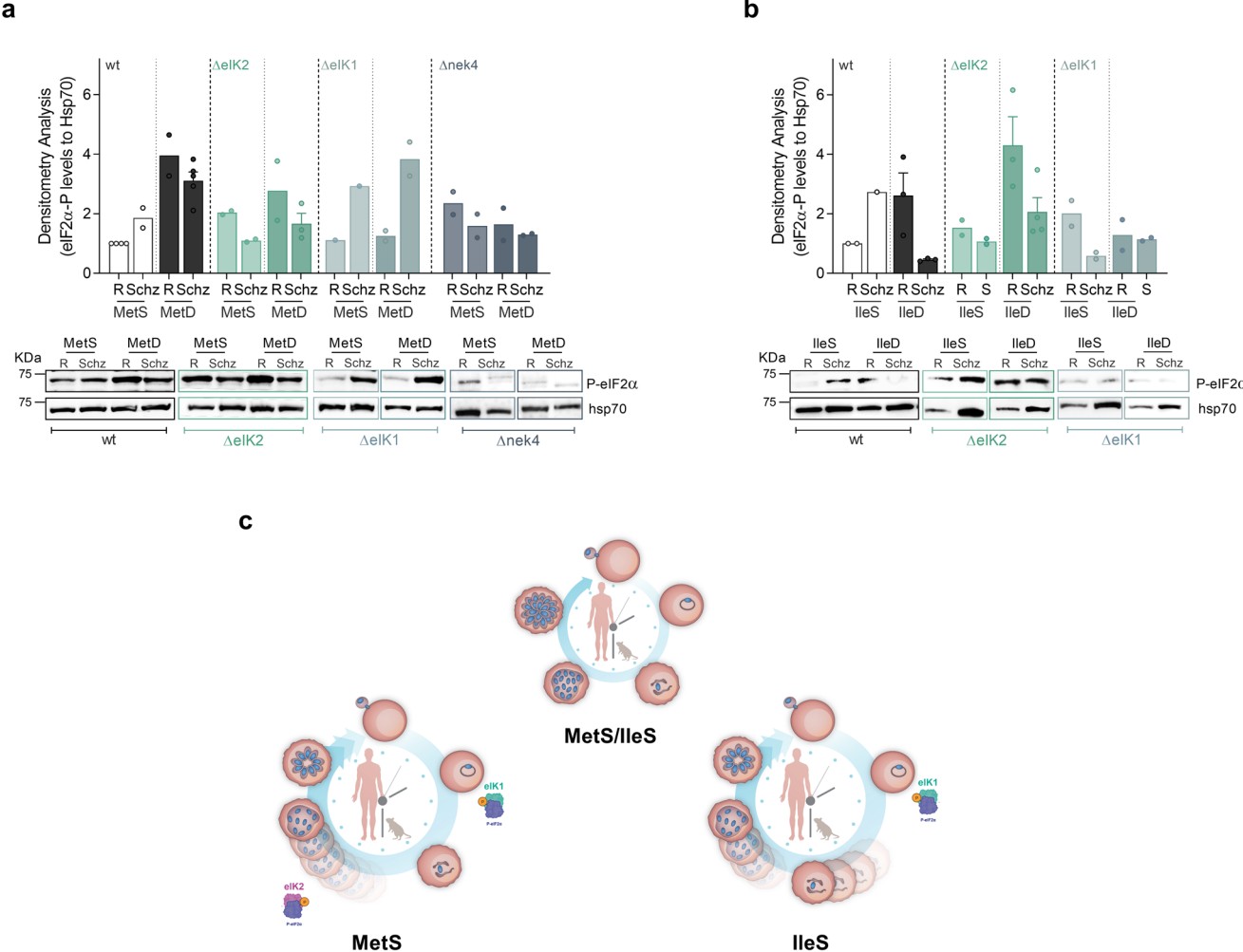

**Fig. 5 Parasite response to AA depletion results in enhanced phosphorylation of the *Plasmodium falciparum* eIF2α ortholog. a, b** Time course analysis of eIF2α phosphorylation status upon AA starvation of sorbitol-synchronized *Pf*3D7 WT, eIK2, eIK1 and nek4 kinase knockout mutants (ring-stage window of ±8 h). Parasites were cultured in (**a**) MetS and MetD media or (**b**) IleS and IleD media. Starvation pulses were performed for 6 h at the ring- (R) or schizont- (Schiz) stage. Parasite proteins were extracted and probed for the phosphorylation of the translation initiation factor 2α (P-eIF2α) at the serine residue 51. Representative phospho-western blots and bar graphs quantifying phosphorylation levels of eIF2α, in ring- and schizont-stage parasites, relative to levels in ring-stage parasites growing under control conditions (MetS or IleS, respectively). Bars represent the mean of eIF2α-P levels relative to hsp70 levels ± SEM. **c** Schematic model for parasite's adaptive response to AA depletion during intraerythrocytic development. *Plasmodium* parasites sense and actively respond to fluctuations in the availability of two distinct AAs: Ile and Met. Such response is heterogeneous as it depends on the AA that is depleted yet, it implies a common pathway comprising the phosphorylation of eIF2α by two distinct kinases, eIK1 and eIK2, which are temporally regulated at distinct stages of the life cycle, allowing parasites to fine-tune replication and development accordingly. This response leads to decreased replication, entry into a hibernation-like state and ultimately, extended survival.

AA that is lacking. Despite the cost to offspring, such an adaptive response ensures that parasites can balance nutrient availability with growth, ultimately extending survival. In contrast, nek4 parasites' inability to control replication and growth under AA deficiency, which is evidenced by the complete loss of eIF2α phosphorylation, results in high growth rates at first but quickly leads to parasite death. Interestingly, ΔeIK1 and ΔeIK2 parasites exhibit an intermediate phenotype under AA deficiency, with each kinase regulating either replication or growth and phosphorylating eIF2α in a stage-specific manner. Despite the ability of eIK2 ring-stage parasites to adapt, schizonts fail to phosphorylate eIF2α in response to AA deficiency, suggesting that parasite replicative rates might be pre-determined at this stage. While eIK1 schizonts can adapt, ring-stages fail to phosphorylate eIF2α in response to AA deficiency. This most likely explains ΔeIK1 parasites' ability to complete the cell cycle and develop into mature schizonts, as eIF2α phosphorylation at the schizont stage

has been shown to be critical for schizont maturation under basal conditions[28].

Interestingly, eIF2α kinases' stage-specific and sequential activation constitutes a major hallmark of the integrated stress response in other eukaryotes[48] including the closely related apicomplexan parasite *Toxoplasma gondii*[49]. Our data show that adaptation to low AA levels entails a common AA-sensing pathway, comprising Nek4 and the phosphorylation of eIF2α by eIK1 or eIK2. However, the overall parasite response to the two AAs tested, Met and Ile, exhibits some peculiarities, with each AA affecting parasite progression through the cell cycle as well as the dynamics of eIF2α phosphorylation in a unique manner. This is not surprising when taking into account current evidence in other eukaryotes showing that GCN2-mediated response to AA starvation differs according to the specific AA that is depleted[50–53]— with Met starvation leading to the most striking decrease in protein translation in mammalian cells[54].

Further studies may be needed to fully appreciate the metabolic rewiring induced by AA starvation and how this triggers the signaling cascade driving the adaptive processes described herein. Still to be revealed is whether Nek4 acts upstream of eIK1 and eIK2, which would support our phenotypic analysis showing a role for Nek4 in both replication and development. Additional studies addressing whether, under AA depleting conditions, eIK1 and eIK2 activation is modulated by Nek4 activity in asexual blood stages would provide further insight on how this AA sensing pathway is regulated in the parasite. Additional studies addressing the role of PK4 (already shown to phosphorylate eIF2α during schizont development[27,28]) in parasite response to AA starvation may also provide additional insight into the dynamics eIF2α phosphorylation and the function of these kinases as critical components of a GCN2-like sensing pathway in the parasite. Moreover, future experiments investigating the phosphorylation status and activity of *Maf1* under AA deficiency, the sole downstream component of the TOR pathway identified to date in *Plasmodium*, may disclose whether a TOR pathway operates in *Plasmodium* and how it interacts with the GCN2 pathway to ensure a suitable cellular adaptation to nutritional fluctuations.

While eradicating malaria is highly desirable, it appears to be rather challenging to achieve. This is primarily due to the emergence of parasite resistance to artemisinin-based combination therapies (ACTs), the forefront treatment for *Plasmodium falciparum* malaria. While the mechanism of action of artemisinin (ART) is not fully understood, ART treatment has been shown to block host cell hemoglobin uptake while markedly increasing general protein damage, resulting in delayed parasite clearance and the persistence of hemoglobin non-degrading ring-stage forms[55]. Therefore, ART treatment is mostly effective at later stages of development when parasites exhibit high metabolic activity[55–57]. Moreover, in addition to a transient reduction in cell cycle pace, ART treatment also enhances eIF2α phosphorylation, thereby resembling the amino acid withdrawal-induced phenotypes. Importantly, increased eIF2α phosphorylation has been shown to be essential for ART-induced latency and parasite recrudescence[58]. Such a programmed state of low metabolic activity, both upon ART-treatment or AA withdrawal, could render parasites insensitive to a broad range of inhibitors, as already shown for the exoerythrocytic stages of other human malaria parasites[59]. By blocking the molecular players identified as part of a *Plasmodium* AA sensing pathway, namely Nek4 or the two eIF2 kinases, eIK1 and eIK2, these findings could represent an alternative and promising strategy to prevent the parasite's ability to enter this latent phase, thereby precluding malaria recrudescence following ART treatment. Furthermore, interventions targeting these kinases would transform highly replicating and as such virulent parasites into attenuated parasite forms, leading to low parasitemia, previously shown to be protective in humans[60].

## Methods

### Chemicals and reagents
Roswell Park Memorial Institute (RPMI) 1640 medium, no glutamine (21870076), HEPES (15630), Gentamicin (15750), L-glutamine (25030-024), Albumax II (11021-029) and SYBR green I (S7567) were purchased from Gibco, Thermo Fisher Scientific. RPMI 1640 Medium, no isoleucine (R9014) was purchased from US Biologicals. Bovine Serum Albumin (BSA) Fraction V (MB046) was purchased from NZYTech. RPMI 1640 no methionine, no cysteine (R7513), Glucose (G6152), Saponin (47036), Spermine (S4264), Spermidine (S0266), Putrescine (D13208), L-Glutathione reduced (G4251), L-Homocysteine (69453) and Trimethoprim (TMP) were purchased from Sigma-Aldrich. S-Adenosylmethione-1,4-butanedisulfonate (NA58198) was purchased from Biosynth Carbosynth. Paraformaldehyde was purchased from Santa Cruz Biotechnology.

### Mice, diets and treatments
Male C57BL/6 J and BALB/c wild-type mice, aged 5–8 weeks, were purchased from Charles River Laboratories (Saint-Germain-sur-l'Arbresle, France). All mice used in this study were housed in the Rodent Facility of Instituto de Medicina Molecular João Lobo Antunes (Lisboa, Portugal), five per cage and kept in specific-pathogen-free conditions. Mice were randomly assigned to different experimental conditions and allowed free access to water and food. Blinding was not possible in MetD-fed mice as mice exhibit a clear difference in body weight. All experimental procedures involving mice were approved by iMM's Animal Welfare Body (ORBEA-iMM), licensed by the national regulator, Direcção Geral de Alimentação e Veterinária (DGAV), and performed in strict compliance with national and European regulations. Methionine- sufficient (MetS) and deficient (MetD) diets were manufactured and purchased from SSniff (Soest, Germany). TMP was provided in drinking water at 0.25 mg per mL starting 2 days prior to infection and changed every 48 h.

### Parasite lines
The following parasite lines were employed in this study: GFP-expressing *P. berghei* ANKA (GFPcon, clone 259cl2), obtained from the Leiden Malaria Research Group; *P. berghei* ANKA kinase deficient parasite lines, were previously generated[37]; *P. falciparum* NF54 and *P. falciparum* 3D7 were obtained through MR4 (www.mr4.org); *P. falciparum sams*-glmS transgenic parasite line was kindly provided by Björn Kafsack (Weill Cornell Medical College, NY, USA) and the *P. falciparum* 3D7 kinase deficient parasite lines were kindly provided by Mathieu Brochet (University of Geneva, Geneva, Switzerland) and Christian Doerig (Monash Biomedicine Discovery Institute, Clayton, Australia).

The *P. berghei* SAMS-DD parasite line was generated by double crossover homologous recombination of the p*Pbsams*.DD.HA construct at the *sams locus* in a GFP expressing parental line (line-507cl1)[61]. The p*Pbsams*.DD.HA construct contains a truncated *Pbsams* ORF in fusion with a destabilizing domain (DD)—a mutant form of the *E. coli* dihydrofolate reductase (ecDHFR) protein, engineered to be degraded[30]—an HA tag and also a cassette for transgenic expression of the human *dhfr*—conferring resistance to pyrimethamine -which is flanked by the *sams* 3'UTR (Supplementary Fig. 1d). Transfection was performed using a protocol described elsewhere[61]. Briefly, blood from a BALB/c mouse infected with the parental line was collected and cultured for 16 h ex vivo. Mature schizonts were purified by a Nycodenz gradient and transfected using Amaxa electroporation system (Lonza). Transfected merozoites were injected into the tail vein of a BALB/c mouse, positively selected by providing pyrimethamine in drinking water (70 µg/ml) and harvested on day 7–10 post transfection. Pyrimethamine resistant, transgenic parasites were dilution cloned before further experimentation and genotyped by PCR (Supplementary Fig. 1e) using the primers listed in Supplementary Table 1.

### Mice infections and in vivo Parasitemia determination
Infection of experimental mice was performed by intraperitoneal (i.p.) inoculation of $1 \times 10^6$ infected RBCs (iRBCs) into C57BL/6 J or BALB/c mice, obtained by passage in the correspondent genetic background mice. For GFP-expressing parasites, peripheral blood parasitemia was determined by flow cytometry analysis of one drop of tail blood collected in 200 µL PBS. A total number of 1 million events (day 1–2 of infection) or 100–200 thousand events (onwards) were acquired on a BD LSR Fortessa using the BD FACSDiva™ Software (v6.2) and the parasitemia expressed as the percentage of GFP+ RBCs. For non-GFP expressing parasite lines, parasitemia was monitored by Giemsa-stained thin blood smears[62].A total of 5–10 thousand RBCs per slide, was randomly acquired and the percentage of infected RBCs (iRBCs)—parasitemia —was semi-automatically counted using Fiji software (version 2.1.0) (https://imagej.net/software/fiji/).

### *P. falciparum* culture and parasitemia determination by flow cytometry analysis
*P. falciparum* NF54, 3D7 and derived knockout clones were maintained in culture[63] at 5% hematocrit in human O + erythrocytes (Instituto Português do Sangue e Transplantação). Of note, whereas the *P. falciparum* NF54 parasite line exhibits a 48 h cell cycle length, the *P. falciparum* 3D7 line shows an accelerated asexual blood stage cycle of 36–42 h. Parasites were cultured in Malaria Complete parasite Media (MCM) prepared by supplementing RPMI 1640 no glutamine, with 2mM L-glutamine, 2 g/L NaHCO₃, 5 mM glucose, 25 mM Hepes, 100 µM hypoxanthine, 10 µg/mL gentamicin, and 10% Albumax II—except for the AA starvation assays. For such experiments, MCM media was prepared by adding all compounds to RMPI lacking either methionine (MetD) or isoleucine (IleD). Respective control media, containing methionine (MetS) or isoleucine (IleS) were prepared by supplementing deficient media with the appropriate AAs, at the concentrations found in regular RPMI 1640 (100 µM Met, 382 µM Ile). Synchronous *P. falciparum* cultures were obtained according to previously described protocols[62,64]. To achieve synchronization, *P. falciparum* parasites were treated twice with 5% sorbitol to achieve a ring-stage window of ±8 h interval in the previous cycle. To do so, cultures were allowed to grow up to 5–10% parasitemia and harvested when parasites were mostly at the ring stage. Parasite pellets were treated with 5% sorbitol for 10 min at RT, centrifuged at 600x g and washed thrice in MCM. Parasitemia was assessed as well as parasite morphology and parasites were subcultured at 5% hematocrit for one additional cycle. The sorbitol treatment was repeated after one cycle—~48 h for NF54 and 40 h for 3D7—as previously described[64]. Ring-stages were then cultured at 2% hematocrit and the starting parasitemia adjusted to 0.5% in 96 well plates containing 200µl of the respective media. Parasitemia in each

developmental cycle was analyzed by flow cytometry analysis after SYBR Green staining[65]. Briefly, 50 μL of *P. falciparum* cultured samples were mixed with the same volume of SYBR Green I solution, previously diluted in water (1:1000). Samples were incubated in the dark for 20 min at room temperature and washed in 500 μL PBS. Samples were acquired on a BD LSR Fortessa using the BD FACS-Diva™ Software (v6.2). A total of 100,000–200,000 events was analyzed per condition. The data was further analyzed on FlowJo X (v10.7.1, TreeStar) using the gating strategy detailed in Supplementary Fig. 4a–c.

**P. berghei ex vivo maturation assays.** *P. berghei* blood-stage parasites were collected by cardiac puncture of BALB/c-infected mice (1–2% parasitemia), washed in pre-warmed incomplete RPMI and cultured in 96-well plates containing complete RPMI culture medium, prepared as follows: RPMI 1640, no glutamine (21870076), supplemented with 5 mM glucose, 2mM L-glutamine, 25 mM HEPES, 2 g/L NaHCO₃, 50 μg/L Gentamicin and 20% of Fetal Bovine Sera (FBS). Homemade complete medium sufficient for methionine (MetS) was prepared by supplementing RPMI 1640 no methionine, no cysteine (R7513) with the respective amino acids at the concentration found in regular RPMI (208 μM and 100 μM, respectively); whereas methionine deficient media (MetD) consisted of RPMI 1640 no methionine, no cysteine (R7513) medium supplemented only with cysteine. Isoleucine-sufficient (IleS) media was prepared by supplementing RPMI 1640 Medium, no isoleucine (IleD; R9014) with the concentration of the respective amino acid found in complete RPMI media (382 μM). Ninety-six-well plates were incubated at 37 °C in an atmosphere of 5% O₂, 5% CO₂ and 90% N₂ for 18–30 h, allowing parasites to mature from rings/young trophozoites into segmented schizonts.

**Quantification of parasite stage progression and merozoite number.** Parasite cell cycle progression was assessed at distinct time points for the rodent malaria parasite, or every 12 h for the human malaria parasite. Parasite staging was performed by microscopic analysis of 1–10 thousand, randomly acquired, Giemsa-stained RBCs. Mature schizonts were assessed by microscopic analysis of Giemsa-stained smears and manually quantified using ImageJ software. Fifty to one hundred segmented schizonts with clearly individualized merozoites containing a single hemozoin crystal were quantified per condition. Additionally, rodent blood-stage nuclei were stained with DAPI and merozoite number was determined by fluorescence microscopy.

**Ex vivo screen of met-downstream metabolites.** To screen for methionine-downstream metabolites, wild-type parasites were cultured in MetS or MetD media and IleS or IleD media and further supplemented with each metabolite—and the appropriate vehicle (as control). The final concentration used for each metabolite was set according to the physiological ranges described during a malaria infection or as previously described elsewhere, as follows: 1.5mM S-adenosylmethionine (SAM)[66], 1 mM glutathione (GSH)[67–69], 20 μM homocysteine (Hcy)[70], 1 mM putrescine (Putr)[71], 1 mM spermine (SpM)[72], and 1 mM spermidine (SpD)[72]. For *P. berghei*, an ex vivo maturation assay was employed with each metabolite being added to the media and incubated for 24- or 30 h. For *P. falciparum*, each metabolite was added to the culture media at day 4 of in vitro culture and incubated throughout one complete intra-erythrocytic developmental cycle. Ninety-six-well plates were incubated at 37 °C in an atmosphere of 5% O₂, 5% CO₂ and 90% N₂. The effect of metabolite supplementation on parasite stage progression and schizont development was assessed by microscopy of Giemsa-stained smears as described above.

**Thin layer chromatography determination of polyamines.** Polyamines were separated by thin-layer chromatography as previously described[73,74]. For all samples, cells were collected and centrifuged. Pellets were washed with Phosphate Buffered Saline (PBS) and then resuspended in 100 μL of 2% perchloric acid. Samples were then incubated overnight at 4 °C. One hundred μL of supernatant was combined with 200 μL of 5 mg/ml dansyl chloride (Sigma Aldrich) in acetone and 100 μL of saturated sodium bicarbonate. Samples were incubated in the dark overnight at room temperature (RT). Excess dansyl chloride was cleared by incubating the reaction with 100 μL 150 mg/mL proline (Sigma Aldrich). Dansylated polyamines were extracted with 50 μL toluene (Sigma Aldrich) and centrifuged. Five μL of sample was added in small spots to the TLC plate (silica gel matrix; Sigma Aldrich) and exposed to ascending chromatography with 1:1 cyclohexane: ethylacetate. Plate was dried and visualized via exposure to UV.

**Amino acid starvation assays for SDS-PAGE analysis.** *P. falciparum* 3D7 parasites and knockout parasite lines, namely the two eIF2α kinases[45], *Pf*ΔeIK1 (PF3D7_1444500)—previously described to closely cluster with mammalian GCN2—and *Pf*ΔeIK2 (PF3D7_0107600)—were maintained as described above in MCM media supplemented with human O + erythrocytes at 2% hematocrit. Parasites were sorbitol-synchronized to the ring-stage and allowed to grow in culture for up to 5% parasitemia. Parasites were collected at two distinct stages of the intra-erythrocytic developmental cycle: ring (12 h) or schizont (38 h). Ring- or schizont-stages were pelleted by centrifugation, washed twice in PBS, pH7.5, equally partitioned and cultured at 37 °C with 5% CO₂ for a 6-h pulse in media

deficient for methionine- or isoleucine- (MetD or IleD) and the respective control media (MetS or IleS)—prepared as previously described. Cultures were then harvested by centrifugation at 1300x g for 5 min at RT and further processed for parasite pellet isolation.

**Parasite pellet harvesting and isolation for SDS-PAGE.** Pelleted erythrocytes were washed in equal volume of PBS, harvested by centrifugation at 1300x g for 2 min at 4 °C and further lysed on ice by treatment with 0.15% saponin in PBS. Parasite pellets were isolated by centrifugation at 5500x g for 10 min, 4 °C and parasites were washed twice in PBS. Blood-stage parasite proteins were isolated by lysis in 50–200 μL of radioimmunoprecipitation assay (RIPA) buffer and washed in PBS with cOmplete™ protease inhibitor cocktail (Roche). Samples were resuspended in 5x SDS-Laemmli buffer and denatured at 95 °C.

**Immunoblotting analysis.** Denatured parasite proteins were resolved on 10% SDS-polyacrylamide gel and transferred to 0.2 μm nitrocellulose membranes (Bio-Rad) using wet-transfer in Towbin buffer (25 mM Tris, 192 mM Glycine (Sigma) and 20% ethanol in ddH2O). Membranes were blocked in 5% BSA in PBS-0.1% Tween 20 (PBST) at RT for 1 h, with rocking. Membranes were then incubated with the appropriate primary antibodies, diluted 1:1000 in 5% BSA in PBST, for overnight at 4 °C. Membranes were washed thrice in 0.1% PBST for 10 min (RT) followed by incubation with appropriate secondary antibodies (conjugated to Horseradish peroxidase) diluted 1:10,000 in 5% BSA in 0.1% PBST, for 60 min at RT. Membranes were washed thrice and bound antibodies were detected by using Immobilon ECL Ultra Western HRP Substrate (Merck Millipore) on the Chemi-Doc XRS + system.

The primary antibodies used for probing membranes were rabbit α HA (clone C29F4, Cell Signaling Technology, Danvers, MA, USA), rabbit α phospho-eIF2α (S51) (119A11, Cell Signaling Technology) and mouse α *Pb*hsp70 (produced in house). The secondary antibodies used were: goat α mouse IgG, HRP conjugate (BML-SA204-0100, Enzo Life Sciences, Lausen, Switzerland) and goat α rabbit IgG, HRP-linked Antibody (7074, Cell Signaling Technology).

**Immunofluorescence assay.** Immunofluorescence analysis of SAMS-DD-HA protein levels was performed in blood smears of *P. berghei* parasites ex vivo cultured for 24 h. MetS-fed BALB/c mice infected with *P. berghei* SAMS-DD parasites were either treated, or not, with TMP in drinking water 2 days prior to infection to achieve SAMS stabilization or knockdown, respectively. Parasites were cultured for 24 h as previously described for the ex vivo maturation assays and blood smears were fixed in 4% PFA-PBS solution for 10 min at RT. Fixed parasites were washed in PBS and permeabilized in 0.1% PBS-Triton X-100 solution for 10 min at RT. Cells were then blocked in 3% BSA for 1 h at RT. Primary antibodies were diluted 1:400 in blocking solution and incubated at 4 °C, overnight. Slides were rinsed thrice in PBS and incubated in secondary antibodies or dyes, diluted 1:400 or 1:1000, respectively, in blocking solution. Stained samples were rinsed in PBS, mounted in Fluoromount G and allowed to dry overnight. Primary antibodies used for fluorescence microscopy include: rabbit α HA (C29F4, Cell Signaling Technology) and mouse α *Pb*hsp70 (produced in house). Secondary antibodies and dyes used for fluorescence microscopy include Alexa-488 conjugated donkey α mouse GFP (A21311, Thermo Fischer Scientific), Alexa-647 conjugated donkey α rabbit IgG (A32795, Thermo Fischer Scientific) and DAPI for nuclear staining (D1306, Invitrogen, Thermo Fischer Scientific). Slides were imaged in Zeiss LSM 710 confocal laser point-scanning fluorescence microscope (ZEN 2 software, Blue edition) and merozoite counting was performed by the use of ×63 oil objective.

**Statistics and reproducibility.** Significance was determined by different statistical tests on the GraphPad (Prism, version 8.4.3) software, as follows: Mann-Whitney U tests was used for comparisons between two different groups or conditions, two-way ANOVA was used to compare parasitemia, development and merozoite numbers between the control counterparts and the following experimental conditions: MetD-, IleD-, kinase mutants or metabolite supplementation assays. The log-rank (Matel-Cox) test was used to compare the survival distributions of two groups. Significance was considered for *p*-values below 0.05. The outliers in the boxplots represent 10% of data points. Biological replicates *(N)* indicated in figure legends refer to the number of parasites. The number of independent experiments is referred to as *n*. Sample sizes were chosen on the basis of historical data. No statistical method was employed to predetermine sample size.

**Reporting summary.** Further information on research design is available in the Nature Portfolio Reporting Summary linked to this article.

## Data availability

All data generated or analyzed during this study are included in this manuscript and its Supplementary Information. Source data underlying the graphs and charts presented in the main and supplementary figures are available as Supplementary Data 1. Full blots are shown in Supplementary Figs. 11–12.

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

## Acknowledgements

From IMM-JLA, Lisbon, Portugal, we would like to acknowledge Inês Bento for discussions, critical input and reading of the manuscript, Ângelo Ferreira Chora for guidance with flow cytometry analysis and for critical reading of the manuscript, Yash G. Pandya for critically reviewing this manuscript, the Bioimaging Facility for assistance with microscopy experiments, the Rodent Facility for overall animal husbandry and support with mice handling and the Flow Cytometry facility for assistance with equipment handling. We would like to thank Mathieu Brochet (University of Geneva Department of Microbiology and Molecular Medicine, Genève, Switzerland) and Christian Doerig (Monash Biomedicine Discovery Institute, Clayton, Australia) for kindly gifting the *P. falciparum* kinase mutant parasite lines: ΔeIK1, ΔeIK2 and Δnek4. We would like also to express our gratitude to Arthur Scherf (Biology of Host-Parasite Interaction, Institut Pasteur, Paris, France) for discussions and for critically reviewing this manuscript. Work at iMM-JLA was supported by Fundação para a Ciência e Tecnologia (PTDC/BIA-MOL/30112/2017) and the "La caixa" Banking Foundation (HR17-00264-PoEMM) grants attributed to V.Z.L. and M.M.M., respectively. Work at the Weill Cornell Medicine (B.F.K. and C.T.H.) was funded by NIH 1R01 AI141965 (B.F.K.), NIH 1R01 AI138499 (B.F.K.), Alice Bohmfalk Charitable Trust Research Grant (B.F.K.), NIH 5F31AI136405-03 (C.T.H.). Work at the Loyola University Chicago (B.C.M. and V.M.) was supported by funds from NIH NIGMS R35GM138199. I.M.M., S.M. and V.Z.-L. were supported by Fundação para a Ciência e Tecnologia, Portugal (PD/BD/135454/2017, DL57/2016/CP1451/CT0010 and DL 57/2016/CP1451/CT0022 respectively).

## Author contributions

I.M.M., V.Z.-L. and M.M.M. conceived and led the study. I.M.M. conducted all the experiments with the assistance of V.Z.-L, S.M. and A.P. V.M. and B.C.M. performed the TLC analysis. C.T.H. and B.F.K. generated the *P. falciparum sams*-glmS transgenic parasite line. O.B. contributed with conceptual input, manuscript reviewing and interpretation. I.M.M. and M.M.M. wrote the manuscript.

## Competing interests

The authors declare no competing interests.
