## [Peer Review File · Communications Biology]

REVIEWERS' COMMENTS:

Reviewer #1 (Remarks to the Author):

This reviewer has read and commented on the manuscript at least three times. Finally, the hard work (also from this reviewer) has paid off. The data are convincing and in support of the main hypothesis. However, two minor issues remain.

1. It would be nice to support the conclusions drawn from Figure 2f by quantitative data. That should not be too demanding, assuming the authors have repeated the experiment several times.
2. I would further suggest to include a statement in the discussion such as: Further studies may be needed to fully appreciate the metabolic rewiring induced by AA starvation and how this triggers the signal cascade leading to the adaptive processes described herein.

Reviewer #2 (Remarks to the Author):

Marreiros et al. have addressed the majority of my recommendations satisfactorily, with changes in the text that improve the clarity of the manuscript and the interpretation of the results.

The parasite response to AA deficiency appears to be complex, with differences between the response to depletion of different amino acids, between parasite stages, several kinases involved at different steps of the regulatory process, and at least two major phenotypes involved. This manuscript is a valuable initial contribution to gain insight into the complexity of this important process.

I only have a few additional minor comments:

-Of the two major phenotypes associated with AA deficiency that are characterized, changes in replication rates and alterations in developmental stage transitions, the former is easy to interpret, but the later is far more complex. Following recommendations from the previous round of review, in the revised manuscript (new text in lines 153-6 and elsewhere), the authors describe that the alterations in developmental stage transitions upon AA depletion likely reflect that a fraction of the parasites undergo growth arrest (similar to hibernation), rather than delayed cycle progression. However, in other parts of the text, the authors still refer to delayed progression/maturation or reduced cell-cycle pace (e.g., lines 139-40, 149, 341). Since these results are complex, to avoid an ambiguous message I recommend to consistently avoid statements that refer to changes in the speed of cycle progression, as in my opinion it is not demonstrated that they occur. All results can be explained by a fraction of the parasites entering hibernation.

-For consistency, panels c and d of Fig. S5 should be moved into the main Fig. 3. These panels present an important result, and for the analogous Pb experiments the results of merozoites per schizont analyses are presented in the main figure (and also in Fig. 1 and 2, results of parasite stage distribution as well as merozoites per schizonts are presented in the main figure).

-Line 303. Add 'previously', when referring to "a kinase only known to function during gametocytogenesis"?

-Lines 342-3. 'Throughout development' applies only to MetD, not IleD, according to Fig. 4? (in wt parasites, increased phosphorylation under IleD conditions is observed only in rings).

-Line 367, 'modulates' or 'is modulated by'? (the data presented suggests that Nek4 operates upstream of eIK1 and eIK2, as described by the authors).

Rebuttal Letter

Reviewers Comments:

Reviewer #1:

It would be nice to support the conclusions drawn from Figure 2f by quantitative data. That should not be too demanding, assuming the authors have repeated the experiment several times.

We thank the reviewer for the suggestion and we have now added this in the new version of the manuscript (Figure 2g).

I would further suggest to include a statement in the discussion such as: Further studies may be needed to fully appreciate the metabolic rewiring induced by AA starvation and how this triggers the signal cascade leading to the adaptive processes described herein.

We agree and acknowledge the reviewer for the suggestion. We have now added this in the new version of the manuscript. (Lines 366, 367; Discussion).

Reviewer #2:

Of the two major phenotypes associated with AA deficiency that are characterized, changes in replication rates and alterations in developmental stage transitions, the former is easy to interpret, but the latter is far more complex. Following recommendations from the previous round of review, in the revised manuscript (new text in lines 153-6 and elsewhere), the authors describe that the alterations in developmental stage transitions upon AA depletion likely reflect that a fraction of the parasites undergo growth arrest (similar to hibernation), rather than delayed cycle progression. However, in other parts of the text, the authors still refer to delayed progression/maturation or reduced cell-cycle pace (e.g., lines 139-40, 149, 341). Since these results are complex, to avoid an ambiguous message I recommend to consistently avoid statements that refer to changes in the speed of cycle progression, as in my opinion it is not demonstrated that they occur. All results can be explained by a fraction of the parasites entering hibernation.

We apologize if this was not clear in the previous version of the manuscript and we thank the reviewer for bringing this to our notice. To minimize uncertainty and misconceptions, we have now updated the revised version of the manuscript. We replaced sentences like "cell cycle arrest" or "delayed cell cycle" or "reduced cell cycle pace" with notions like "entry into a hibernation-like state" (Lines 195, 219, 271, 342, 748) or "impaired development" (Lines 130, 150, 205, 208, etc.).

For consistency, panels c and d of Fig. S5 should be moved into the main Fig. 3. These panels present an important result, and for the analogous Pb experiments the results of merozoites per schizont analyses are presented in the main figure (and also in Fig. 1 and 2, results of parasite stage distribution as well as merozoites per schizonts are presented in the main figure).

We agree and thank the reviewer for the suggestion. In order to include panels c and d of Figure S5 in the main figures, we created a new figure (Figure 4 in the revised version of the manuscript) that now includes both parasitemia and merozoite counts per schizont in the human malaria parasite *P. falciparum*. The same results for the analogous *P. berghei* are presented in main Figure 3 of the revised version of the manuscript.

Line 303. Add ‘previously’, when referring to “a kinase only known to function during gametocytogenesis”?

We thank the reviewer for the suggestion and we have now added it in the revised version of the manuscript (Line 306).

Lines 342-3. ‘Throughout development’ applies only to MetD, not IleD, according to Fig. 4? (in wt parasites, increased phosphorylation under IleD conditions is observed only in rings).

We apologize if this was not clear in the previous version of the manuscript and we have changed this sentence to accommodate the reviewer’s suggestion as follows: “*an adaptive mechanism that requires enhanced phosphorylation of eIF2 α , which must either be maintained throughout development (MetD) or not (IleD), depending on the AA that is lacking. Despite the cost to offspring, such an adaptive response ensures that parasites can balance nutrient availability with growth, ultimately extending survival*” (Lines 343-346).

Line 367, ‘modulates’ or ‘is modulated by’? (the data presented suggests that Nek4 operates upstream of eIK1 and eIK2, as described by the authors).

We thank the reviewer for bringing this to our attention. We have now corrected this sentence in the new version of the manuscript, as follows: “*eIK1 and eIK2 activation is modulated by Nek4 activity*” (Line 370).